# The Indices of Cardiovascular Magnetic Resonance Derived Atrial Dynamics May Improve the Contemporary Risk Stratification Algorithms in Children with Hypertrophic Cardiomyopathy

**DOI:** 10.3390/jcm10040650

**Published:** 2021-02-08

**Authors:** Lidia Ziółkowska, Łukasz Mazurkiewicz, Joanna Petryka, Monika Kowalczyk-Domagała, Agnieszka Boruc, Katarzyna Bieganowska, Elżbieta Ciara, Dorota Piekutowska-Abramczuk, Mateusz Śpiewak, Jolanta Miśko, Magdalena Marczak, Grażyna Brzezińska-Rajszys

**Affiliations:** 1Department of Cardiology, The Children’s Memorial Health Institute, 04-730 Warsaw, Poland; l.ziolkowska@ipczd.pl (L.Z.); m.kowalczyk@ipczd.pl (M.K.-D.); a.boruc@ipczd.pl (A.B.); k.bieganowska@ipczd.pl (K.B.); g.brzezinska@ipczd.pl (G.B.-R.); 2CMR Unit, Department of Cardiomyopathies, National Institute of Cardiology, 04-628 Warsaw, Poland; 3CMR Unit, Department of Coronary and Structural Heart Diseases, National Institute of Cardiology, 04-628 Warsaw, Poland; jpetryka@ikard.pl; 4Department of Medical Genetics, The Children’s Memorial Health Institute, 04-628 Warsaw, Poland; e.ciara@ipczd.pl (E.C.); d.piekutowska@ipczd.pl (D.P.-A.); 5CMR Unit, National Institute of Cardiology, 04-628 Warsaw, Poland; mspiewak@ikard.pl (M.Ś.); jmisko@wp.pl (J.M.); mmarczak@wp.pl (M.M.)

**Keywords:** hypertrophic cardiomyopathy, atrial strain, outcome, risk markers

## Abstract

Introduction: The most efficient risk stratification algorithms are expected to deliver robust and indefectible identification of high-risk children with hypertrophic cardiomyopathy (HCM). Here we compare algorithms for risk stratification in primary prevention in HCM children and investigate whether novel indices of biatrial performance improve these algorithms. Methods and Results: The endpoints were defined as sudden cardiac death, resuscitated cardiac arrest, or appropriate implantable cardioverter-defibrillator discharge. We examined the prognostic utility of classic American College of Cardiology/American Heart Association (ACC/AHA) risk factors, the novel HCM Risk-Kids score and the combination of these with indices of biatrial dynamics. The study consisted of 55 HCM children (mean age 12.5 ± 4.6 years, 69.1% males); seven had endpoints (four deaths, three appropriate ICD discharges). A strong trend (DeLong *p* = 0.08) was observed towards better endpoint identification performance of the HCM Risk-Kids Model compared to the ACC/AHA strategy. Adding the atrial conduit function component significantly improved the prediction capabilities of the AHA/ACC Model (DeLong *p* = 0.01) and HCM Risk-Kids algorithm (DeLong *p* = 0.04). Conclusions: The new HCM Risk-Kids individualised algorithm and score was capable of identifying high-risk children with very good accuracy. The inclusion of one of the atrial dynamic indices improved both risk stratification strategies.

## 1. Introduction

The grimmest consequence of hypertrophic cardiomyopathy (HCM) is arrhythmic sudden cardiac death (SCD). The electric instability arises mainly from disturbed fibre architecture, vast fibrosis and microvasculature imperfections in hypertrophied hearts [1,2,3,4]. Reliable identification of patients endangered by lethal arrhythmias remains a cornerstone of the modern assessment of patients with HCM [5,6]. However, the search for the perfect algorithm seems to be an endless one; it needs to be relatively easy-to-use, incredibly precise and remarkably reliable. Contemporary risk stratification methods in adults and children are based on a combination of multiple outcome markers [7] or mathematical algorithms that provide individualised risk score [8]. The recently introduced HCM Risk-Kids stratification model is the first model specially designed and validated for children [9]. However, it was retrospectively built, no randomised trials were performed and no prospective prediction models were constructed to guide appropriate implantable cardioverter-defibrillator (ICD) implantation in childhood HCM. Moreover, this new strategy was not tested against an operational algorithm based on the general expert consensus that only proven risk factors are useful predictors of SCD in the paediatric population.

Similar to the European Society of Cardiology (ESC) Risk Score for adults, the new HCM Risk-Kids equation also contains the left atrial (LA) size index as a predictive outcome parameter. This consolidated LA role came from several studies that found that the dimensions and function of the systemic atrium were modulated by multiple factors, such as left ventricle compliance, fibrosis and hypertrophy [10,11,12,13]. Thus, LA size was found to be a sensitive marker of LV function, which might provide additional data for disease staging and earlier detection of disease progression than conventional functional LV measures [10,12]. Advances in cardiac magnetic resonance (CMR) derived feature tracking (FT) technology has enabled the study of the advanced mechanical components of atrial function. The determinants and magnitude of LA malfunction were previously identified in adults [13] and children [14] with HCM. However, the usefulness of CMR-derived LA mechanical indices in risk stratification models have not been validated.

Therefore, in this outcome study, we aim to compare the abilities of the proven American College of Cardiology/American Heart Association (ACC/AHA) strategy and the newly implemented HCM Risk-Kids algorithm in identifying children at-risk. We also investigate whether modern LA mechanical indices might improve the predictive performance of these operational risk stratification models.

## 2. Materials and Methods

### 2.1. Study Population

The study was approved by the Institutional Ethics Committee and written informed consent to participate in the study was obtained from all subjects and their parents.

The study cohort comprised 55 consecutive, prospectively recruited children with HCM, 19 of whom had left ventricle outflow tract obstruction (LVOT) obstruction. Criteria for inclusion in the study were age <18 years old at the time of diagnosis and echocardiographic evidence of LV hypertrophy defined as a diastolic septal thickness or LV diastolic wall thickness *z*-score > 2 [determined as more than two standard deviations from the mean value for the population corrected for body surface area (BSA)] in the absence of any other cardiac or systemic disease capable of producing a similar amount of hypertrophy. No children needing sedation during CMR scan were enrolled in this study. Patients with either paroxysmal or persistent atrial fibrillation were excluded from the study.

### 2.2. Follow-Up

Study entry was either the first hospital or outpatient visit in our HCM tertiary referral centre for children. The clinical/survival status was archived during hospital follow-up and outpatient visits or telephone contact with children’s parents. Follow-up time was calculated from study entry to most recent contact or endpoint. No patient was lost to follow-up. The primary endpoint was the composite of all-cause death, resuscitated cardiac arrest due to severe ventricular arrhythmia and appropriate ICD discharge.

### 2.3. Risk Stratification

Risk stratification was performed for each child using a conventional strategy, which is a part of ACC/AHA guidelines [4] for HCM, and by calculating the HCM Risk-Kids risk score [5].

The ACC/AHA risk markers were family history of SCD attributable to HCM in ≥1 first- or second-degree relatives, recent unexplained syncope, massive LV hypertrophy (maximal wall thickness ≥ 30 mm or a *z*-score ≥ 6), nonsustained ventricular tachycardia and hypotensive or blunted blood pressure response to exercise.

The HCM Risk-Kids score, which is the probability of SCD at 5 years, was expressed in percentage and calculated using the following equation: (1)$$1-0.949437808^{\;\exp\left(\mathrm{Prognostic}\;\mathrm{Index}\right)}$$
where Prognostic Index = 0.2171364 × (MWT *z* score − 11.09) − 0.0047562 × (MWT *z*-score^2^ − 174.12) + 0.130365 × (LA diameter *z*-score − 1.92) + 0.429624 × unexplained syncope + 0.1861694 × non sustained ventricular tachycardia − 0.0065555 × (maximal left ventricle outflow tract gradient − 21.8).

### 2.4. CMR Examination and Image Analysis

Standard CMR study was performed using a 1.5 Tesla scanner (Sonata and Avanto, Siemens, Erlangen, Germany). A stack of short-axis breath-hold steady-state free precession (SSFP) images from base to apex (typical imaging parameters: repetition time, 2.2 to 3.6 ms; echo time, 1.2 ms; flip angle, 64 to 79 degrees; slice thickness, 8 mm; gap 2 mm) was obtained.

SSFP images served for the calculation of ventricular volumes and ejection fraction with the use of dedicated software (MASS 7.6, Medis, Leiden, The Netherlands). Manual delineation of endocardial and epicardial contours was performed in end-diastolic and end-systolic phases. LV end-diastolic volume (LVEDV), LV end-systolic volume (LVESV), LV mass (LVM) and left ventricular ejection fraction (LVEF) were calculated. LVEDV, LVESV and LVM were indexed to BSA (LVEDVI, LVESVI and LVMI, respectively). The papillary muscles were excluded from the LV mass calculation.

LA and RA volumes were quantified using dedicated software (MASS 7.6, Medis, Leiden, The Netherlands). LA volumes were calculated according to the biplane area-length method. Manual tracking of the LA area and length were performed in the 2- and 4-chamber views, excluding pulmonary veins and the LA appendage. RA volumes were calculated according to the single plane area-length method. Manual tracking of RA area and length was performed in the 4-chamber view. LA volumes, indexed for body surface area (BSA) were assessed at LV end-systole (LAV max), at LV diastole before LA contraction (LAV pac) and at late LV diastole after LA contraction (LAV min). RA volumes, indexed for BSA, were assessed at RV end-systole (RAV max), at RV diastole before RA contraction (RAV pac) and at late RV diastole after RA contraction (RAV min). LA and RA volumetric analyses were performed twice by two independent and skilled observers. Total atrial emptying fraction (LAEF total, RAEF total corresponding to LA and RA reservoir, respectively), passive atrial emptying fraction (LAEF passive, RAEF passive corresponding to LA and RA conduit function, respectively) and an active atrial emptying fraction (LAEF booster, RAEF booster corresponding to LA and RA contractile booster pump function, respectively) were defined according to the following formulas:EFtotal = (Vmax − Vmin) × 100/Vmax(2)
EFpassive = (Vmax − Vpac) × 100/Vmax(3)
EFbooster = (Vpac − Vmin) × 100/Vpac(4)

### 2.5. Feature Tracking Analysis

Atrial strains and strain rates were analysed using dedicated software (CVI42, Calgary, AB, Canada). Left atrial endocardial borders were tracked in the 2- and 4-chamber views. Right atrial borders were tracked in the 4-chamber view. The atrial endocardial border was manually delineated in the diastolic phase and tracked automatically. Tracking performance was visually reviewed to ensure accurate tracking of the atrial tissue. Manual adjustments were made in case of inaccurate atrial tracking. If the tracking quality was insufficient due to the presence of pulmonary veins or left atrial appendage, the corresponding segment was excluded from the analysis. Atrial feature tracking (FT) analysis was performed twice by two experienced observers. We analysed three aspects of the LA and RA mechanics: passive strain (εE, corresponding to atrial conduit function), active strain (εA, corresponding to atrial contractile booster pump function) and total strain, as well as the sum of passive and active strains (εS, corresponding to atrial reservoir function). Three strain rate parameters were evaluated: peak positive strain rate (SRs, corresponding to atrial reservoir function), peak early negative strain rate (SRe, corresponding to atrial conduit function) and peak late negative strain rate (SRa, corresponding to atrial contractile booster pump function).

### 2.6. Echocardiographic Examination

Two-dimensional, conventional pulsed Doppler and M-mode echocardiography was performed at rest using standard methods (ultrasound machine iE 33, Philips, Healthcare). Conventional pulsed Doppler was used to record the mitral regurgitation in the apical 4-chamber view and marked visually in 4 (1-trace, 2-small, 3-moderate, 4-large) grade scale.

To determine maximal degree of LVOTO two-dimensional and Doppler echocardiography was performed during the Valsalva manoeuvre in the sitting position, and then during standing if no gradient was provoked. The maximum gradient greater than 30 mmHg was considered significant. Neither pharmacological provocative nor exercise tests were used to determine maximal LVOT gradients. Furthermore, CMR cine images were visually assessed for the presence of LVOTO.

### 2.7. Statistics

All of the continuous variables were expressed as mean ± standard deviation (SD) or as the median and interquartile range (IQ range) and were tested for normality using the Kolmogorov–Smirnov test. Comparisons between groups were performed using the Student’s *t*-test or the Wilcoxon-Mann-Whitney *U* test for continuous variables and the chi-squared or Fisher’s exact test for categorical variables, as appropriate.

We built and compared several logistic regression models in three steps to determine which combination of candidate predictors had the best capability in identifying endangered children. At the baseline step, we constructed and compared two models: (1) the ACC/AHA Model, which involves an analysis of one or more major, conventional risk marker according to the ACC/AHA consensus guidelines (family history of SCD, syncope, massive LV hypertrophy, nonsustained ventricular tachycardia and abnormal blood pressure response to exercise); and (2) the new HCM Risk-Kids Model, which involves an analysis of risk score estimated using a mathematical algorithm.

In the second step, we constructed and compared 12 separate models that were created by adding every volumetric and contractile index of both the left and right atrium functions to the AHA/ACC and new HCM Risk-Kids models.

In the final third step, we compared first step Models (ACC/AHA, new HCM Risk-Kids) with second step Models (containing various LA performance indices).

The ability of each model to identify children with endpoints was estimated using the area under (AUC) the receiver operating characteristic (ROC) curve with the Youden index. The AUCs for each model were compared using the DeLong method. The model fit was assessed using the Hosmer–Lemeshow goodness-of-fit test.

The net re-classification improvement was quantified as a sum of differences in proportions of individuals moving up, minus the proportion moving down, for people who develop events and as the proportion of individuals moving down, minus the proportion moving up for people who do not develop events. The significance of the re-classification was tested using the McNemar test.

A two-sided *p*-value less than 0.05 was considered to indicate statistical significance. Statistical analyses were performed using MedCalc 12.1.4.0 software (MedCalc, Mariakerke, Belgium).

## 3. Results

The comparisons between baseline clinical, CMR and FT markers in HCM patients and healthy controls were presented elsewhere [14].

The baseline clinical and CMR comparisons between children with and without endpoints are presented in Table 1. Children with endpoints did not differ from the non-endpoint subjects in terms of any anatomical and functional parameters. In total, 15 (27.2%) children underwent genetic testing. Among these, in eight patients (53%), we found mutations in genes with known correlation with HCM (Table 2). Among subjects with mutations, one (12.5%) child had an endpoint. In this case, we detected the *TNNI3* (troponin I3, cardiac type; NM_000363.4) variation c.557G>A leading to missense substitution p.(Arg186Gln). The variant was already reported as pathogenic in hypertrophic cardiomyopathy (RCV000167988.8) and primary familial hypertrophic cardiomyopathy (RCV000157533.4, RCV001258033.1).

Figure 1 and Figure 2 contain the comparisons of biatrial FT analysis for children with and without endpoints. Only LA conduit and reservoir strain indices, and RA conduit strain rate were significantly worse in children with endpoints.

### 3.1. Outcome

The median of follow-up time was 7.91 years (IQR, 7.31–8.40). The average number of major risk factors was 1.11. We had a full set of prognostic data, except for the results of the exercise test, which was available for only 18 patients; blunted blood pressure reaction was observed in no subjects. The Appendix A contains summary of all risk factors in all patients.

There were 41 (74.5%) children with one and more risk factors. The median of HCM Risk-Kids score was 4.24% (IQR, 2.31–7.76).

Seven endpoints occurred during the follow-up. Four children died, all of them suddenly. Of these, three patients died before systemic employment of ICDs in primary SCD prophylaxis. No patients declined the recommended ICD therapy. Also, there were three appropriate ICD interventions. Table 3 gives the predictive outcome characteristics of children with endpoints, including baseline classic risk markers, all data needed to calculate the HCM Risk-Kids score, and the HCM Risk-Kids score itself.

### 3.2. Risk Prediction Models

The ACC/AHA model yielded satisfactory performance (AUC = 0.638; 95% CI, 0.496–0.765) in identifying children who experienced HCM complications. The HCM Risk-Kids Model achieved good performance in identifying endangered subjects (AUC = 0.724; 95% CI, 0.569–0.824). There was a strong trend towards better prediction capabilities of the Risk-Kids Model relative to the ACC/AHA Model (DeLong *p* = 0.08) (Figure 3).

Based on the ROC curve analysis, we estimated that the optimal cut-off point for the HCM Risk-Kids risk score for prediction of events was >3.73% [AUC = 0.728; 95% CI—0.590–0.840]. At this threshold, the sensitivity was 100%, and specificity was 47.8%. Clinical outcomes for patients with new HCM Risk-Kids score > 3.73% were significantly worse than for those with a score < 3.73 (log-rank *p* = 0.01) (Figure 4).

### 3.3. Role of Atrial Dynamics in Risk Stratification

Next, we constructed several regression models incorporating novel components of left atrial function to both the HCM Risk-Kids and ACC/AHA Models. Adding LAεe (corresponding to atrial conduit function) significantly increased the prediction performance of the AHA/ACC Model (to an AUC of 0.859; 95% CI, 0.757–0.949; DeLong *p* = 0.01 when comparing the native and enriched ACC/AHA Models) and the HCM Risk-Kids algorithm (to AUC of 0.867; 95% CI, 0.748–0.949; DeLong *p* = 0.04 when comparing the native and enriched HCM Risk-Kids Models). There were no differences in prediction abilities between ACC/AHA and HCM Risk-Kids modified strategies (DeLong *p* = ns) (Figure 5).

Using ROC curve analysis, we estimated the optimal cut-off point for LAεe for prediction of the event at <7.95% (AUC = 0.859; 95% CI, 0.737–0.938). At this threshold, the sensitivity was 100%, and specificity was 71.7%. Clinical outcomes for patients with LAεe < 7.95% were significantly better than for those with LAεe > 7.95% (log-rank *p* < 0.01).

The other LA or RA indices (mechanical, volumetric or contractile) did not improve the predictive capabilities of the risk stratification algorithms. The discriminating performances of all enriched models are summarised in Appendix A. The predictive performance of presence of LVOTO, LVOT gradient and mitral regurgitation were low and significantly lower (*p* = ns) than investigated algorithms (Appendix A). Also, the Kaplan–Meyer curves showed no differences in survival between children with and without LVOTO (*p* = ns) (Appendix A).

### 3.4. Reclassifications

The native ACC/AHA algorithm classified 41 children as high risk, of which seven (17.1%) were correct, using the optimal cut-off point for HCM Risk-Kids score (>3.73%), 33 children were identified as high risk, of which seven were correct (21.2%) (McNemar *p* = 0.02).

Also, children were categorised into subgroups based on the presence of ACC/AHA classic risk factors and values of LAεe: HIGH RISK > 1 ACC/AHA risk factors and LAεe < 7.95%; LOW RISK, the rest of the study group with no ACC/AHA risk factors OR LAεe > 7.95%. This strategy classified 20 children as high risk, of which seven were correct (35%). The clinical outcomes for the HIGH-RISK subjects, based on modified ACC/AHA classification, were worse than for the LOW-RISK subjects (log-rank *p* < 0.01).

Finally, we categorised the study population into subgroups based on the values of HCM Risk-Kids scores and values of LAεa: HIGH RISK, HCM Risk-Kids score >3.73% and LAεa < 7.95%; LOW RISK, the rest of the study group with HCM Risk-Kids score <3.73% or LAεa < 7.95%. This enriched HCM Risk-Kids strategy classified 18 children as high risk, of which seven (38.8%) had events. The clinical outcomes for the HIGH-RISK subjects, based on the modified HCM Risk-Kids classification, were worse than in the LOW RISK subjects (log-rank *p* < 0.01).

There were no differences in re-classification accountabilities when comparing the modified ACC/AHA and HCM Risk-Kids strategies (McNemar *p* = ns).

The modification of both algorithms did not improve by the net number of correctly classified children (high risk). However, incorporating the functional parameters of LA greatly reduced the number of incorrectly identified subject (false-positive results). Both modified re-classifications based on enriched ACC/AHA and HCM Risk-Kids strategies were significantly better than the native counterparts (McNemar test between native and novel AHA/ACC and Risk-Kids Models *p* < 0.01 for both comparisons).

## 4. Discussion

This outcome study aimed to evaluate the prognostic abilities of novel markers of LA dynamics. We also compared the predictive performance of both traditional AHA/ACC risk stratification strategy and the new HCM Risk-Kids algorithm with the LA dynamic indices enriched models for arrhythmic events in the paediatric population with HCM. Our main findings can be summarised as follows:There was strong trend towards better prediction capabilities of the Risk-Kids Model compared to ACC/AHA ModelIncorporating one of the left atrial displacement markers improved the accuracy of both methods for the prediction of fatal cardiac events.The enriched ACC/AHA and new HCM Risk-Kids Models performed similarly at identifying at-risk HCM children.

### 4.1. Atrial function

The left atrial function is an essential feature of cardiovascular performance; it physiologically comprises three different phasic components [15]. The atria serve both as a reservoir and as a conduit for collecting and conveying the blood volume from the pulmonary and systemic veins to respective ventricles.

Additionally, atrial ejection force might deliver up to 30% of LV filling and cardiac output [16]. LA size has long been considered a marker of chronic diastolic dysfunction, effectively predicting cardiovascular morbidity and death. LA mechanics, however, which were never included into routine clinical imaging, seem to allow a complete evaluation of cardiovascular performance, representing a more sensitive method of detecting subclinical changes of diastolic function [17,18].

Previous, mainly echocardiographic, studies reported that left atrial function and size are modulated mainly by LV diastolic function and performance [16,18]. In adult patients with HCM, the substrates for diastolic dysfunction, represented mainly by hypertrophy and fibrosis, were associated with LA enlargement and functional abnormalities [13].

The reports regarding RA dynamics are very sparse. No papers describe CMR-derived studies on juvenile subjects with HCM. Authors of the existing works, mostly based on pulmonary hypertension models, accordingly conclude that the decline in compliance of right ventricle is the presumable mechanism of deterioration of RA function [19,20]. Also, not without significance, is backwardly transmitted, through pulmonary circulation, elevated left-ventricular end-diastolic pressure which may additionally decrease RA function [14]. Moreover, the progressive nature of fibrosis resulted from primary muscle disease can contribute to further deterioration of RA size and function. All these make one assume that the reasons of RA malfunction in HCM children is surely multifactorial and need to be investigated in future works.

### 4.2. Baseline Prediction Models

Our study delivers head-to-head comparisons of contemporary risk stratification strategies in a purely paediatric population. We show that the new HCM Risk-Kids algorithm was nearly better than the traditional ACC/AHA risk markers. These results are surprising and challenge the notion that only traditional HCM risk markers are useful predictors in young HCM patients, especially given that both models share three common indices. However, the new HCM Risk-Kids algorithm heavily relies on continuous variables and the assumption that their escalation is associated with continuous change in risk prediction [9]. Interestingly, Norrish at al. found that the LVOT gradient appears to be inversely associated with the risk of SCD [9]. In our cohort, clinical outcomes were not significantly worse in children with LVOT obstruction. However, it was previously demonstrated that left-ventricular [21] and biatrial performances [14] were far worse in subjects with LVOT obstruction. The loud murmur produced by obstruction might result in early diagnosis during the routine paediatric examination, which could lead to closer follow up and early-introduced pharmacological treatment.

The inclusion of LA size in the algorithm represents a novel approach, for which the predominant mode of demise in HCM patients is progressive diastolic dysfunction leading to advanced heart failure [8,22,23]. Interestingly, the ESC Risk score, which also contains the maximal LA dimension, was found to be highly ineffective in identifying high-risk children, adolescence and young adults [22]. However, the adult HCM Risk Score included the absolute values of LA dimension and LV wall thickness, which can be inappropriate for smaller patients. It is possible that the inclusion of child-specific z-scores contributed to the great improvement in prediction capabilities of the HCM Risk-Kids Model. Nevertheless, some authors suggest that the volume of LA seems to have superior value over LA size as a prognostic marker in adult patients with HCM. On the other hand, Kowallick and colleagues found that minimum LA volumes were more closely associated with LV fibrosis than hypertrophy [13]. This marker might provide additional data since it is measured in end-diastole after being directly exposed to LV end-diastolic pressure. In our study, neither minimal nor maximal LA volume provided improved accuracy in identifying high-risk children compared to any other volumetric feature of LA. Moreover, none of these (maximal, minimal, or pre-contraction) improved the prediction capabilities of the risk stratification models. Presumably, minor substrates for increased LA size such as diastolic dysfunction, hypertrophy, fibrosis or mitral regurgitation underscores the value of these LA components in paediatric population. Moreover, it was previously demonstrated that changes in LA function precede changes in LA size in the pathogenesis of HCM [24].

It should be remembered that both native ACC/AHA and new HCM Risk-Kids Models considerably overestimate the number of high-risk children. On the other hand, due to the progressive nature of HCM, more extended observations are needed to verify the proportion of children who would develop life-threatening events in adolescence and adulthood. Nevertheless, our results suggest that traditional, binary risk factors are not able to assure risk prediction performance, unlike adequately weighted and age-calibrated traditional and functional risk markers might.

### 4.3. Left Atrial Mechanical Indices Predictive Capabilities

Several studies in adult HCM subjects have shown that LA global longitudinal strain (GLS), measured by speckle tracking echocardiography, was associated with clinical outcome [25,26,27]. However, none of these works was performed in a paediatric population. In our population, the reservoir component (which seem to be the counterpart of atrial GLS in echo) was not a useful predictor of arrhythmic events. It can be speculated whether different strain measurement methodologies, limited echocardiographic visibility or much younger age of our patients impacted the results.

Among the various markers of LA mechanics incorporated into the HCM Risk-Kids algorithm, the conduit components emerged as predictors, which might considerably improve the discriminative abilities of the model. Strain analysis is a valuable diagnostic tool because it might inform on functional LA remodelling more specifically and earlier then volumetric assessment only [24]. Although LA dilatation could also be a physiological response to compensate for decreased LA function, LA strain analysis reveals intrinsic LA dysfunction and LA stiffness at an early stage [25,26]. In healthy, asymptomatic subjects, LA conduit function is responsible for early diastolic LV augmentation and probably has a predominant contribution to cardiovascular performance [15]. Therefore, it can be easily compromised by early diastolic LV abnormalities, such as LV stiffness [28]. Also, LA conduit function was found to be significantly impaired in heart failure patients with preserved LV systolic function and associated with exercise intolerance [29,30]. On the other hand, several studies have shown that LA conduit function was only partially associated with LV stiffness or relaxation [31,32]. LA conduit dysfunction is considered to be an early sign of LA malfunction and is an early detectable marker in children with less advanced diastolic dysfunction. It was postulated that compromised LA conduit function represents an intrinsic LA pathology and/or fibrosis that cannot be fully explained by co-existing ventricular malfunction [32,33]. The structural LA dysfunction surely originates from the same, mainly in-born, mechanisms, which trigger ventricular hypertrophy. Further studies, including advanced atrial tissue characterisation techniques, are now needed to confirm this thesis.

### 4.4. Limitations of the Study

The limitations of this study are mainly inherent to its design. This was a single-centre study with a relatively small sample size. The results of this study will need to be confirmed in large-scale, multicentre, longitudinal studies. Moreover, our cohort was a part of the original population on which the new HCM Risk-Kids algorithm was modelled. However, the original, multicentre HCM Risk-Kids study contained incomplete data (34% of LA z-scores, 15% of LVOT gradients, 15% of nsVT data were missing and complete data were available for only 51.5% of patients). In contrast, our work comprises a full set of all functional and prognostic data. This makes our study an honest attempt to enrich the work by Norrish et al.

Also, in recent work, Maron et al. propose enhanced ACC/AHA risk markers designed and tested for the adult HCM population but not validated for younger cohorts [34]. None of our patients had any of these enhanced risk markers, such as vast fibrosis (more than 15% of LV mass) or compromised LV systolic function and apical aneurysm. Therefore, Maron’s strategy was not included in our analysis.

## 5. Conclusions

This study confirms that a new risk stratification strategy using the HCM Risk-Kids individualised score is capable of identifying high-risk children with rewarding accuracy. The inclusion of one of the atrial dynamic indices improved both risk stratification strategies and delivered excellent performance in identifying HCM children that subsequently had events.

## Figures and Tables

**Figure 1 jcm-10-00650-f001:**
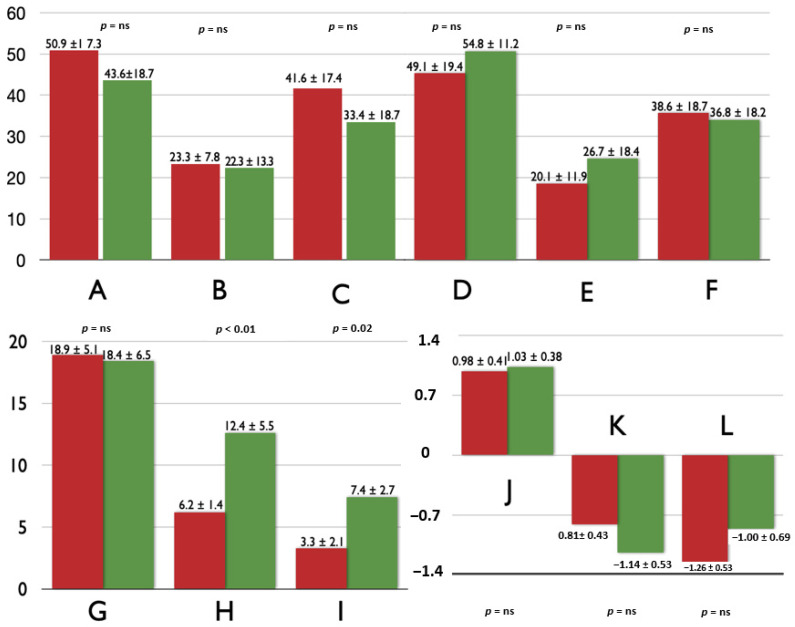
The comparison of left-atrial volumetric [A—indexed maximal volume (mL/m^2^), B—indexed minimal volume (mL/m^2^) and C—indexed volume just before contraction (mL/m^2^)], contractile [D—total emptying fraction (%), E—conduit emptying fraction (%) and F—contractile emptying fraction (%)] and mechanical [G—total strain (%), H—conduit strain (%), I—contractile strain (%), J—total strain rate (1/s), K—conduit strain rate (1/s), L—contractile strain rate (1/s)] components between children with (red) and without (green) endpoints.

**Figure 2 jcm-10-00650-f002:**
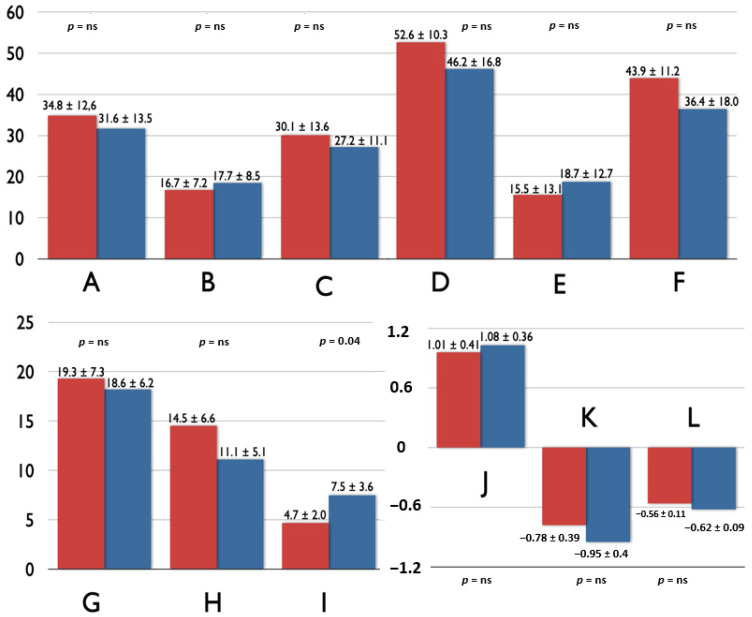
The comparison of right-atrial volumetric [A—indexed maximal volume (mL/m^2^), B—indexed minimal volume (mL/m^2^) and C—indexed volume just before contraction (mL/m^2^)], contractile [D—total emptying fraction (%), E—conduit emptying fraction (%) and F—contractile emptying fraction (%)] and mechanical [G—total strain (%), H—conduit strain (%), I—contractile strain (%), J—total strain rate (1/s), K—conduit strain rate (1/s), L—contractile strain rate (1/s)] components between children with (red) and without (blue) endpoints.

**Figure 3 jcm-10-00650-f003:**
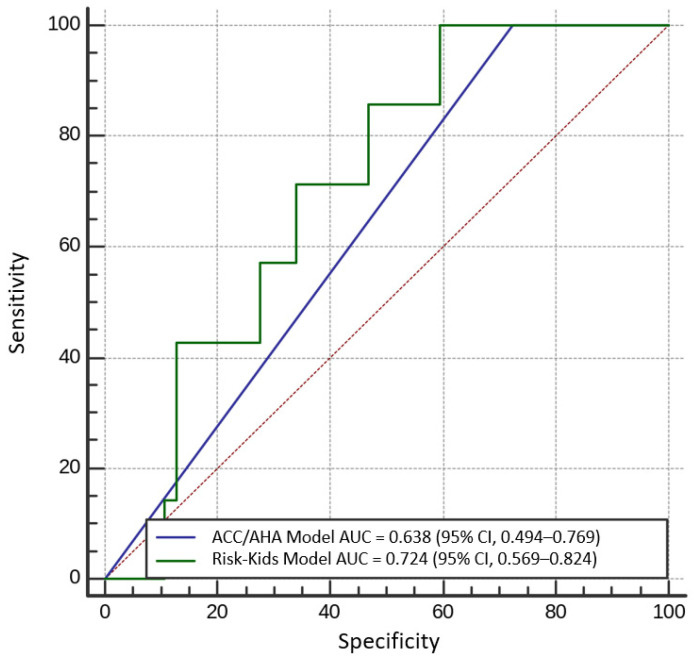
Comparison of receiver operating characteristic curves for baseline ACC/AHA (blue) and new Risk-Kids (green) risk stratification strategies; deLong *p* = 0.08.

**Figure 4 jcm-10-00650-f004:**
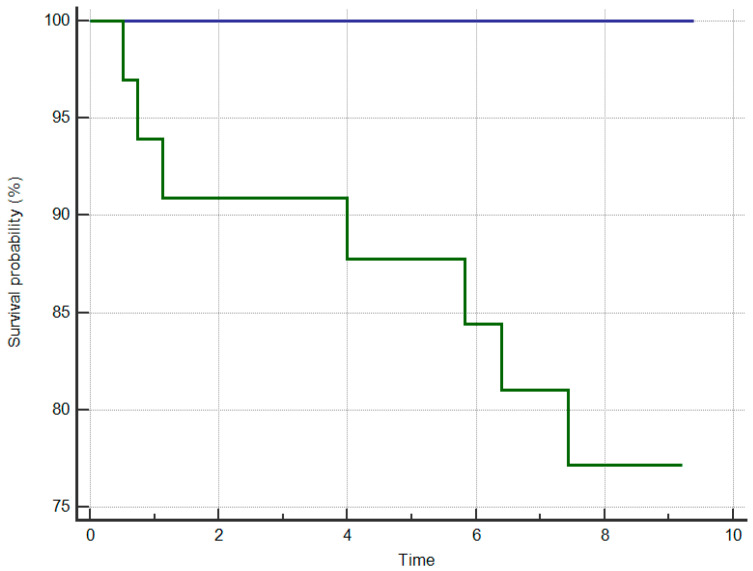
Kaplan-Meier survival curves for patients with new Risk-Kids score >3.73% (green) and <3.73% (blue); log rank *p* < 0.01.

**Figure 5 jcm-10-00650-f005:**
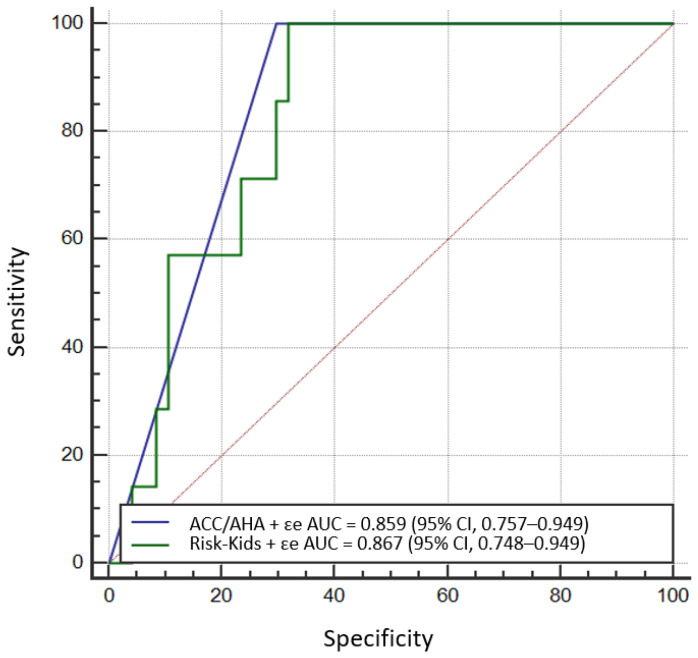
Comparison of receiver operating characteristic (ROC) curves for enriched ACC/AHA (blue) and Risk-Kids (green) Models; deLong *p* = ns. LAεe – Left atrial passive strain corresponding to atrial conduit function.

**Table 1 jcm-10-00650-t001:** Baseline demographic, clinical, volumetric data in whole hypertrophic cardiomyopathy (HCM) cohort and in children with and without endpoints.

	Study Group *n* = 55	Patients with Endpoints *n* = 7	Patients without Endpoints *n* = 48	*p* Value between Children with and without Endpoints
Age (years)	12.5 ± 4.6	10.4 ± 5.4	12.3 ± 4.5	ns
Male sex *n* (%)	38 (69.1)			
BSA (m^2^)	1.5 ± 0.2	1.1 ± 0.4	1.3 ± 0.4	ns
NYHA	1.7 ± 0.5	1.8 ± 0.5	1.7 ± 0.5	ns
IM	1.73 ± 0.53	1.78 ± 0.66	1.71 ± 0.48	ns
LVOTO *n*(%)	19 (34.5)	2 (28.5)	17 (35.4)	ns
MRI parameters				
LVEDVI (mL/m^2^)	79.7 ± 17.5	78.8 ± 14.8	79.8 ± 17.9	ns
LVESVI (mL/m^2^)	27.2 ± 10.1	27.8 ± 10.5	25.3 ± 6.8	ns
LVSVI (mL/m^2^)	52.5 ± 11.5	52.4 ± 11.9	53.7 ± 10.3	ns
LVEF (%)	65.8 ± 7.4	65.7 ± 7.8	68 ± 6.1	ns
LVMI (g/m2)	94.9 ± 59.7	92.1 ± 58.6	110 ± 67.7	ns
Areas of lge (%)	27 (49.1)	8 (100%)	19 (39.5%)	0.02
RVEDVI (mL/m^2^)	73.4 ± 13.5	96.3 ± 46.6	90.3 ± 48.1	ns
RVESVI (mL/m^2^)	27.7 ± 11.2	38.1 ± 22.2	26.3 ± 12.4	ns
RVSVI (mL/m^2^)	43.9 ± 12.2			ns
RVEF (%)	61.4 ± 8.7	60.1 ± 7.2	64.0 ± 7.0	ns

BSA—body surface area; LVEDVI—left ventricle end diastolic volume index; LVESVI—left ventricle end systolic volume index; LVSVI—left ventricle stroke volume index; LVEF—left ventricle ejection fraction; MR—mitral regurgitation; RVEDVI—right ventricle end diastolic volume index; RVESVI—right ventricle end systolic volume index; RVSVI—right ventricle stroke volume index; RVEF—right ventricle ejection fraction; IM—mitral regurgitation; LVOTO—left ventricular outflow tract obstruction.

**Table 2 jcm-10-00650-t002:** The results of genetic testing in all study cohort and in subjects with endpoints.

	All Patients (*n* = 55)	Patients with Endpoints (*n* = 7)
Genetic testing (%)	15 (27.2)	1 (14.2)
*KRAS*	1	0
*MYBPC3*	2	0
*RYR2*	1	0
*MYH7*	2	0
*MYPN*	1	0
*TNNI3*	1	1

**Table 3 jcm-10-00650-t003:** The outcome predictive characteristics of children with endpoints with baseline classic risk markers, all ingredients needed to calculate the Risk-Kids score and the Risk-Kids score.

Patient	Type of Event	Number of ACC/AHA Risk Factors	Risk-Kids Score	Family History of SCD	Massive LV Hypertrophy	Syncope	nsVT	Maximal Wall Thickness *z*-Score	Age at Initial Evaluation	LA Size *z*-Score
W-A	death	1	10.5	0	0	0	1	13.7	13.8	5.1
P-B	death	1	6.01	1	0	0	0	10	18.1	2.3
P-P	death	2	10.2	1	0	0	1	16.3	2.65	2.1
N-K	death	1	10.7	0	0	0	0	15.1	3.69	2.4
N-Ko	ICD dis	2	4.3	1	0	1	0	5.2	13.06	1.2
W-S	ICD dis	1	6.5	0	0	0	0	11.1	7.35	1.4
J-W	ICD dis	1	5.4	1	0	0	0	6.2	15.13	9.4

SCD—sudden cardia death; LV—left-ventricular; nsVT—non sustained ventricular tachycardia; LA—left atrium; ICD dis—ICD discharge.

## Data Availability

The data presented in this study are available on request from the corresponding author.

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
