# Peer review of "The Indices of Cardiovascular Magnetic Resonance Derived Atrial Dynamics May Improve the Contemporary Risk Stratification Algorithms in Children with Hypertrophic Cardiomyopathy"

_jcm, 2021, doi:10.3390/jcm10040650_

Round 1

Reviewer 1 Report

In this manuscript, the authors examined the prognostic utility of the ACC/AHA risk factors, the novel HCM Risk-Kids score, and the combination of these indices with biatrial dynamics in 55 children with HCM. There was a trend for better end-point identification in the HCM Risk-Kids model compared to the AHA/ACC model. Atrial conduit function was found to improve the predictive capabilities of both models.

Overall, the manuscript is well written and a thoughtful attempt to try to add to our current knowledge of risk stratification in a pediatric HCM. However, I do have some questions/comments for the authors:

  1. Most importantly, I think there should be a statistical review of this manuscript. Given the conclusion that the HCM Risk-Kids score is “almost” better than the ACC/AHA risk factors I think it would be important to see if the study was adequately powered to detect a difference in these models based off of the sample size and event rate. Similarly for the atrial volumes and mechanics, since the majority of the values were not different between patient with and without events, was the study adequately powered to detect a difference?
  2. I also question the duration of follow-up of the patients for detecting events. What is the median, IQR and range of follow-up for the subjects? Was there a minimum follow-up time necessary for a subject to be in the study? I think there is danger of saying that children were inaccurately classified if they did not have event during the follow-up period of the study, as many could have future adverse event with longer follow-up.
  3. Were there any children with adverse events who were missed by either of the models?
  4. The numbers on the bar graphs in Figure 1 and 2 are not uniform (some on bars, some above etc). The figures should be cleaned up with consistent labeling.
  5. There is a spelling error in legend of Figure 1 and 2 – without is spelled “without”
  6. I think readers may be interested to see the risk factors of all of the patients (even those without events). Consider including this information as a supplemental table.
  7. Were Inter- and intra-observer reproducibility performed for the atrial measurements and atrial mechanics? This would be of interest as there is some scrutiny regarding the reproducibility of FT strain by MRI.
  8. Since RA volumes and mechanics were investigated, it would be nice to add some discussion of other literature assessing RA function in pediatric HCM patients.

Author Response

All reviewers need to be thanked for the careful reading of our manuscript and their valuable comments which we would be delighted to answer to.

Reviewer #1 

Comment 1

Most importantly, I think there should be a statistical review of this manuscript. Given the conclusion that the HCM Risk-Kids score is “almost” better than the ACC/AHA risk factors I think it would be important to see if the study was adequately powered to detect a difference in these models based off of the sample size and event rate. Similarly for the atrial volumes and mechanics, since the majority of the values were not different between patient with and without events, was the study adequately powered to detect a difference?

Answer for comment 1

The estimated sample size calculated at the study design stage was 51 patients for comparison of ROC curves, 50 subjects for McNemar test and 52 children for comparison of means (all for power 0,8). However, it is generally accepted that post-hoc power analysis is not an appropriate approach once you have done the study. Using estimated effect size in post hoc analysis may provide inadequate estimate of power, and in many cases can lead to drastic distortion of power especially in studies that happen to achieve statistical significance. The statisticians advise to present the 95% CI confidence interval with p-value to show the uncertainty of the results instead of a retrospective power result. We found our work as a preliminary study which for the first time determines indices of atrial dynamics as a predictive data. Moreover, the small sample size was mentioned in Limitation of the study section followed by the information that large, multicenter studies need to validate our findings.

The reviewer is right that term “almost” is not specific enough and was deleted from conclusions.

However, whenever there is any doubt in power estimation calculations statistical review should be performed.

Comment 2

I also question the duration of follow-up of the patients for detecting events. What is the median, IQR and range of follow-up for the subjects? Was there a minimum follow-up time necessary for a subject to be in the study? I think there is danger of saying that children were inaccurately classified if they did not have event during the follow-up period of the study, as many could have future adverse event with longer follow-up.

Answer for comment 2

The median of follow-up time was 7,91 years (IQR – 7,31-8,40). The shortest observation period for patients without event was 5,27 years. The reviewer is right that longer observation time may improve the diagnostic performance of investigated factors. However, our follow-up time seem to be one of the longest in published literature regarding HCM children; the follow-up time in Norrish study based on which the Risk-Kids calculator was developed was more than 2 years shorter (5,3 years).

Comment 3

Were there any children with adverse events who were missed by either of the models?

Answer for comment 3

All children with endpoints were correctly classified as high-risk by both native and modified AHA/ACC and Risk-Kids algorithms. There was no child who were missed by either models. The detailed baseline accountabilities and reclassifications can be found in Reclassification section.

Comment 4

The numbers on the bar graphs in Figure 1 and 2 are not uniform (some on bars, some above etc). The figures should be cleaned up with consistent labeling.

Answer for comment 4

According to reviewer suggestion the Figure 1 and 2 were reformatted and improved.

Comment 5

There is a spelling error in legend of Figure 1 and 2 – without is spelled “without”

Answer for comment 5

This mistake was corrected.

Comment 6

I think readers may be interested to see the risk factors of all of the patients (even those without events). Consider including this information as a supplemental table.

Answer for comment 6

We include Additional Table with all the risk factors in all subjects.

Comment 7

Were Inter- and intra-observer reproducibility performed for the atrial measurements and atrial mechanics? This would be of interest as there is some scrutiny regarding the reproducibility of FT strain by MRI.

Answer for comment 7

The reproducibility analysis was performed and can be found here: doi: 10.1007/s00330-018-5519-7. PMID: 29882072

Comment 8

Since RA volumes and mechanics were investigated, it would be nice to add some discussion of other literature assessing RA function in pediatric HCM patients.

Answer for comment 8

The short paragraph on RA mechanics in children were added into Discussion paragraph:

"The reports regarding RA dynamics are very sparse. No papers describe CMR-derived studies on juvenile subjects with HCM. Authors of the existing works, mostly based on pulmonary hypertension models, accordingly conclude that the decline in compliance of right ventricle is the presumable mechanism of deterioration of RA function. Also, not without significance, is backwardly transmitted, through pulmonary circulation, elevated left-ventricular end-diastolic pressure which may additionally decrease RA function. What is more, the progressive nature of fibrosis resulted from primary muscle disease can contribute to further deterioration of RA size and function. All these make one assume that the reasons of RA malfunction in HCM children is surely multifactorial and need to be investigated in future works"

Reviewer 2 Report

The Authors of the manuscript found that left atrial (LA) function indices assessed by cardiovascular magnetic resonance (CMR) improve the prediction of lethal events over the conventional ACC/AHA strategy in children with Hypertrophic Cardiomyopathy (HCM). The study is interesting and adds new information in the challenging field of sudden-death prediction in children with HCM. The Authors should address the following issues:

  1. The study population prospectively included a consecutive series of 55 HCM-children undergoing CMR. Was CMR included systematically in the diagnostic work-up of HCM at the Authors’ Institution? If this was the case, how many children were excluded from the study due to CMR findings of HCM phenocopies? How many patients were excluded due to previous ICD implantation?
  2. Left atrial size and function indices are recognized as indirect signs of left ventricular diastolic dysfunction. However, HCM patients may show a varying impairment due to stable or fluctuant LV obstruction. The left atrial burden can be quite different in patients with or without LV obstruction. In patients with SAM-related obstruction, the amount of correlated mitral regurgitation may categorize different forms of left-atrial impairment and correlated adverse events. The Authors should re-categorize the patient subgroup according to LV obstruction and the severity of related mitral regurgitation
  3. In HCM patients, left atrial function may also be impaired by primary atrial myopathy. Analysis of fibrosis of the left atrial wall could have provided further information for risk stratification, concomitant with or independent of the hemodynamic burden.

Author Response

All reviewers need to be thanked for the careful reading of our manuscript and their valuable comments which we would be delighted to answer to.

Reviewer #2

Comment 1

The study population prospectively included a consecutive series of 55 HCM-children undergoing CMR. Was CMR included systematically in the diagnostic work-up of HCM at the Authors’ Institution? If this was the case, how many children were excluded from the study due to CMR findings of HCM phenocopies? How many patients were excluded due to previous ICD implantation?

Answer to comment 1

The CMR evaluation is routinely utilized in our institution for evaluation patients with congenital heart diseases and cardiomyopathies including hypertrophic. The investigated cohort of 55 children was initially recruited for our research grant. There were no children excluded from this study based on CMR findings or due to any device implantation.

Comment 2

Left atrial size and function indices are recognized as indirect signs of left ventricular diastolic dysfunction. However, HCM patients may show a varying impairment due to stable or fluctuant LV obstruction. The left atrial burden can be quite different in patients with or without LV obstruction. In patients with SAM-related obstruction, the amount of correlated mitral regurgitation may categorize different forms of left-atrial impairment and correlated adverse events. The Authors should re-categorize the patient subgroup according to LV obstruction and the severity of related mitral regurgitation.

Answer to comment 2

We previously demonstrated that ventricular and atrial dynamics in children are mostly associated with LVOT obstruction. However, in adults Kowallick et al demonstrated the LA dynamics can be linked with LV fibrosis. On the other hand, it appears the LVOT gradient is inversely associated with risk in children with HCM. That finding was underlined by Norrish et al based on recent, large pediatric series. We performed further statistical analysis which can be found in Supplementary Materials; the predictive performance of presence of LVOTO, LVOT gradient and mitral regurgitation, analyzed as single prognostic factors, were low and significantly lower than investigated algorithms. Moreover, the Kaplan-Meyer curves showed no differences in survival between children with and without LVOTO. Therefore, although LVOTO is associated with the atrial dynamics it cannot replace the additional prognostic value of atrial mechanical parameters.

Surely, all the interaction between degree of LVOTO, survival and atrial dynamics are not sufficiently explored and need to be studied in future studies. We are afraid that the further reclassifications may disturb the concept and design of this study and deliver distorted and the incompressible results.

Comment 3

In HCM patients, left atrial function may also be impaired by primary atrial myopathy. Analysis of fibrosis of the left atrial wall could have provided further information for risk stratification, concomitant with or independent of the hemodynamic burden.

Answer to comment 3

The dedicated sequences for atrial fibrosis assessment was not performed in our study cohort. The existing LGE images, especially in small children, are not diagnostic enough the credibly evaluate atrial scar.

Round 2

Reviewer 1 Report

I think this is an interesting study which is well designed and presented. The findings add to the literature trying to predict outcomes in a very challenging disease and lay groundwork for future larger, multicenter studies. There are minor formatting issues: 1) spacing with n(%) is not uniform in Supplemental Table 1. 2) the legend covers up part of the Kaplan Meier curve on a few graphs.

Reviewer 2 Report

The Authors have appropriately changed the manuscript.